# Sickness Presence among Teachers, Nurses and Private Sector Office Workers

**DOI:** 10.3390/healthcare11040512

**Published:** 2023-02-09

**Authors:** Dominik Olejniczak, Agata Olearczyk, Katarzyna Swakowska, Anna Staniszewska, Karolina Zakrzewska

**Affiliations:** 1Chair and Department of Public Health, Medical University of Warsaw, 02-091 Warsaw, Poland; 2AMEOS Klinikum Anklam, Hospitalstrasse 19, 17389 Anklam, Germany; 3Department of Experimental and Clinical Pharmacology, Medical University of Warsaw, 02-091 Warsaw, Poland; 4Department of Epidemiology and Biostatistics, Medical University of Warsaw, 02-091 Warsaw, Poland

**Keywords:** sickness presence, presenteeism, disease, workplace, public health, absenteeism

## Abstract

Introduction: Sickness presence is used to denote an employee who feels unwell but still attends work, thus avoiding absence. The intention of this paper is to compare sickness presence in a group of the following professions: teachers, nurses and private sector office workers. Material and methods: For the purpose of this study, a survey based on the original PAPI form (*Paper-and-Pen Personal Interview*) was carried out. Non-probability sampling, the snowball method (N = 507: teachers *n* = 174, nurses *n* = 165 and private sector office workers *n* = 168), covering the whole of Poland, was adopted. Non-parametric hypotheses were verified using the chi-squared test with a statistical significance α = 0.05. Results: Compared to nurses and private sector office workers, teachers more frequently attended work when sick (*p* < 0.05). Out of the reported ailments that respondents worked with, teachers more often indicated rhinitis (*p* < 0.05), sore throat and cough (*p* < 0.05) and increased temperature (*p* < 0.05). This may be associated with a threat to the health of individuals in their charge. Teachers commonly complained about joint and bone pain (*p* < 0.05) and gastrointestinal disorders (*p* < 0.05). Contrary to nurses and private sector office workers, teachers did not point to ‘lack of a replacement’ as the reason for their presence at work when sick (*p* < 0.05). Exclusively, teachers added financial issues and difficulties in access to healthcare if they are working fewer hours to the list of reasons for attending work when sick. Conclusions: Results suggest that there is a need for further studies on the presence of sick employees in the workplace, especially for teachers. The sickness presence of teachers and nurses may be a threat from a public health perspective. The workplace itself is a significant place to prevent many diseases.

## 1. Introduction

Sickness presence is defined as the situations in which: (1) an employee attends work when sick [1,2] and/or (2) an employee feels unwell, presents ailments and should stay at home (sick leave) but still works [3,4,5]. According to data from Statista, the average level of absenteeism in European countries amounted to 9.9% of working time in the third quarter of 2022 [6]. The estimation of sickness presence is possible using well-known indicators, such as the lost time indicator, frequency of absenteeism and Bradford’s rate [7]. One of the systematic reviews showed that sickness absence in overweight and/or obese employees generates higher costs of absence than in employees with normal weight [8]. In Poland, the number and amount of benefits paid due to sickness absence are constantly increasing [9].

Sickness presence is part of a wider phenomenon referred to as presenteeism. Presenteeism is identified as an ineffective presence in the workplace, a decrease in productivity resulting from health status (both physical and psychological dimensions) or occupational, personal, social and emotional problems [10]. Presence in the workplace despite feeling unwell and presenting with ailments impedes full engagement and functioning at work [10,11]. It results in decreased productivity, distraction, a higher risk of making mistakes and reduced quality of occupational life [12,13]. Sickness presence may be an important public health issue, especially in the case of infectious diseases, as the presence of sick individuals in the workplace poses a threat also for the health of other people (colleagues, individuals in charge and clients) through a risk of infection transmission [14].

Both absenteeism and presenteeism are well-known concepts in economic terms; however, presenteeism is more difficult to assess in terms of its level as well as its impact on costs. One of the methods of assessing the presenteeism rate can be employees’ self-evaluation of their productivity [15]. However, this is a subjective assessment as one’s productivity can be affected by many factors, not only sickness. Other studies suggest that the cost of presenteeism might be five to ten times higher than sickness absence alone [16,17]. Regardless of the scale, the issue of presenteeism and its costs comes up whenever the cost of sickness absence is discussed and, related to employees’ health, they are a significant challenge for employers.

The objectives of this article are to (1) compare the presence of employees in the workplace despite diseases and ailments in a group of teachers, nurses and private sector office workers; (2) analyse the intensity of sickness presence in the aforesaid professions; (3) identify the reasons why they stay at work when sick; and (4) recognize the type of ailments they experience while attending work when sick. This article extends the scope of previous pilot studies [18].

## 2. Material and Methods

A survey based on the original PAPI form (*Paper-and-Pen Personal Interview*) was carried out. Sickness presence was defined as the presence of an employee in the workplace despite a disease or ailments. Two questions concerning sickness presence were asked:Direct question: Does it happen that you attend work when sick (e.g., with a cold, toothache, backache or migraine)? (1) yes, often; (2) yes, sometimes; (3) no, never.Indirect question: Please mark the ailments that you worked with (more than one answer is possible): rhinitis/sore throat, cough/headache/increased temperature/gastrointestinal disorders/toothache/back or neck or joint pain/fatigue/concentration problems.

Respondents were selected on the basis of non-probability sampling. Having used the snowball method, the questionnaire was given to individuals working as teachers, nurses and private sector office workers with a request to give it to persons with the same profession. Representatives of the discussed professions from the whole of Poland were enrolled in the study (N = 507) including:A total of 174 teachers working in primary, middle and secondary schools;A total of 165 nurses working in public or private health care units;A total of 168 private sector office workers.

The characteristic of respondents is presented in Table 1.

A descriptive analysis was performed by calculating the basic statistics (mean, standard deviation and median) of the quantitative variables or counts and percentages for qualitative variables. Non-parametric hypotheses were verified using the chi-squared test or the Fisher exact test, whenever necessary for any comparison, and a *p*-value of <0.05 was considered statistically significant. Statistical analyses were performed using Statistica 10 (licence held by the Medical University of Warsaw).

Females predominated in the group of respondents. The mean age of the study sample, including teachers and nurses, was more than 40 years old. Compared to office workers (~28 years old), this value was higher. Teachers and nurses had employment contracts, while in a group of private sector office workers, civil law contracts were dominant, especially commission contracts. Above all, the respondents worked on a regular working schedule (fixed start and end time of the work). In a group of nurses, there was a substantial percentage of individuals working in the shift system. Teachers and private sector workers worked irregular working times (unexpected end time of work or overtime).

To a large extent, the respondents perceived their health status as good and very good. Long-term health problems or diseases diagnosed were less often reported by nurses and more frequently by teachers (nearly 38% of teachers presented with long-term health problems or diagnosed chronic diseases, in which bone and joint-related problems predominated). Full characteristics of respondents is included in Table 1.

## 3. Results

Nearly all of the respondents attended work when sick or suffering from symptoms, which is presented in Figure 1 and Table 2.

Irrespective of sex, age, having children, health status, health-related problems or profession held, sickness presence affected almost all the respondents.

All respondents—even those who declared that they have never attended work when sick—marked symptoms they worked with, which is presented in Table 3.

Out of the reported ailments that the respondents worked with, teachers more frequently marked rhinitis (chi-squared test = 45.9; *p* < 0.05). All private sector office workers and the majority of teachers worked with a sore throat and a cough (chi-squared test = 120.2; *p* < 0.05). Office workers more commonly attended work with an increased temperature (chi-squared test = 49.3; *p* < 0.05). They also often experienced fatigue (chi-squared test = 11.6; *p* < 0.05), concentration problems (chi-squared test = 45.5; *p* < 0.05) and headache (*p* > 0.05). They also more commonly indicated working despite a toothache (chi-squared test = 15.4; *p* < 0.05). Teachers more frequently indicated gastrointestinal disorders (chi-squared test = 18.3; *p* < 0.05) and bone and joint pain (chi-squared test = 24.3; *p* < 0.05). A substantial majority of nurses worked with headaches and fatigue. Compared to other professions, nurses less frequently attended work when presenting with symptoms of infectious diseases.

It is important to identify the reasons why employees do not stay at home when sick, feel unwell or present health problems, but instead they attend work. The distribution of responses concerning the aforementioned reasons is presented in Table 4.

Out of the reasons for sickness presence, teachers more frequently marked ‘obligations to be met’ (chi-squared test = 36.3; *p* < 0.05) and ‘forthcoming deadline’.

(chi-squared test = 129.1; *p* < 0.05). Exclusively, nurses pointed out a ‘fear’ of employers

(chi-squared test = 127.6; *p* < 0.05). A lack of a replacement was the most common reason in the case of office workers (chi-squared test = 149.3; *p* < 0.05). Contrary to nurses and private sector office workers, teachers did not indicate a ‘lack of a replacement’ as a reason for their presence at work when sick. For some respondents (1⁄5 of nurses, 1/4 of office workers and 1/3 of teachers), loyalty to the institution they work in, employer and colleagues was the reason for sickness presence. Exclusively, teachers signalled financial issues and difficulties in access to healthcare if there are fewer working hours as the reason of attending work when sick.

## 4. Discussion

Presenteeism—a situation in which an employee does not stay at home when sick, experiencing health or personal problems but attends work (e.g., due to a problem providing care for a child), thus working ineffectively—is a new concept in the scientific literature. The first references to presenteeism appeared in the 1990s of the 20th century; however, this phenomenon was present in the working environment prior to paying attention to absenteeism, i.e., the absence of employees from the workplace [4,19]. Despite the long presence of presenteeism in the labour economy, there is no specific model for its assessment, which makes it an even greater challenge for employers to manage.

This case study in Polish illustrates that the concept of presenteeism poses difficulties, in terms of semantics, when attempting to apply it to the Polish language. Thus, the following notions are used: (1) Polonizing the English word ‘presenteeism’ (PL: ‘prezenteizm’) [8,9,13]; (2) ‘ineffective presence at work’ [20]; (3) ‘sickness presence’ to describe presenteeism caused by disease or health problems [13]; and (4) ‘extra presence’ in the workplace [13]. It is possible to confuse presenteeism with presentism (according to the Polish Language Dictionary, presentism is a trend in the modern methodology of historical studies which negates the possibility of learning the objective historical truth; a method of interpreting history by analysing the problems and phenomena of the present times by the past) [13].

Sickness presence is a phenomenon that is difficult to measure as it is seemingly invisible—an employee is present in the workplace; however, he/she suffers from ailments and feels unwell. In the case of absenteeism, the measurement is easier—the lack of an employee is equal to the lack of productivity, and the confirmation of absenteeism is a sick leave issued by a doctor—therefore, we can easily calculate the absenteeism rate using one of the previously described patterns. It is not so obvious for sickness presence, although some studies suggest that the cost of presenteeism can be five or even ten times higher than absenteeism [15,16] because the employee is present in the workplace, however, due to different factors (physical and psychological health status, occupational, personal and emotional problems), he/she cannot fully engage in his/her work. Presenteeism, especially sickness presence, however, cannot be referred only to a decrease in employee productivity or the estimation of costs borne by the employer as a result of the presence of employees in the workplace despite presenting with health problems. Regardless of the scale, the risk of presenteeism and its costs and consequences arise whenever the issue of sick absenteeism is discussed and constitutes a significant challenge for employers and public health. From a public health perspective, it is important to identify:The intensity of sickness presence—who is affected and how often.The reasons for which employees attend work despite suffering from symptoms.The ailments and health problems that accompany sickness presence.

Such an approach provides a possibility for a decent analysis of presenteeism (it was also observed by *Agata Wężyk* and *Dorota Merecz* [18]). Therefore, to analyse sickness presence in this study, the authors used a survey based on employee self-evaluation.

Sickness presence affected nearly all of the respondents whether or not they were teachers, nurses or private sector office workers. It affected females and males of all age groups, despite of having children or not. Those who considered their health as good or bad, as well as those who reported long-term health problems or not, attended work when sick.

The analysis of sickness presence is associated with methodological problems. Absenteeism may be measured on the basis of data on employee absence and its reasons (sickness absence reported by a sick leave note issued by a doctor, taking care of a child reported by the employee themself). Such data are collected by the employers as well as, e.g., by the Social Insurance Institution. The analysis of sickness presence is mainly restricted to surveys. Authors often use a direct question with a request to indicate how often the health status of the employee limits his/her functioning to such an extent that he/she would benefit from an absence from work/sick leave, but he/she still goes to work [4,5]. In the present study, respondents were requested to indicate how frequently they go to work when sick. Additionally, an indirect question was asked in which employees were to indicate the symptoms they work with. In this study, all respondents, even those who declared that they did not attend work when sick, marked symptoms they worked with. The discrepancy between a declared sickness presence (direct question) and actual presence in the workplace when sick (marking the ailments the respondents worked with) was reported in the previous study [17].

The fact that respondents work with symptoms of infectious diseases raises concerns (rhinitis, sore throat, cough and increased temperature). They may be a source of infection to their colleagues, individuals in their charge or clients, which is what we observed during the recent COVID-19 pandemic or during the annual increase in flu incidence. It is especially dangerous in the case of nurses and teachers. Of importance is that the results of this study show that teachers more frequently attend work despite presenting with symptoms of upper respiratory tract infection. Studies suggest that sickness absence due to influenza decreases virus transmission in the workplace even by 39% [14]. Thus, health education of employees, especially those who have close contact with individuals with impaired immunity (patients, children and the elderly) is an important issue. A few days of sick leave may alleviate the acute ailments and lead to a full presence at work, while several days of sickness presence with an infectious disease may exacerbate health problems (leading to complications) and pose a threat to the health of contacts. It is of importance to use the workplace as a key place to raise the awareness about this issue both among employees and employers. It is also relevant to work on policy which does not accept the presence of employee with symptoms of infectious diseases in the workplace.

Other diseases of increasing importance in the context of sickness absenteeism and presenteeism are non-communicable diseases, such as diabetes and hypertension [21]. From year to year, we can observe increasing incidence of chronic and so-called civilization diseases and related sick leaves. According to data from the European Agency for Safety and Health at Work and the Central Institute for Labour Protection, in most developed countries, diseases of the musculoskeletal system and mental disorders are the leading causes of sick leave [22]. In 2020, the Social Insurance Institution of Poland paid almost PLN 23 million in sickness allowances [23].

An employee’s illness has wide-ranging consequences. These are significant costs for the budget due to paid benefits (sickness and disability allowances) as well as costs to the health care system. For the employee, this is also a direct cost in terms of spending for medicines and medical care outside the public system, as well as lower income. It is also a cost for the employer due to the employee’s sickness absence and presenteeism. Interestingly, the employees themselves show interest in preventive initiatives aimed at civilization and oncological diseases as a part of occupational healthcare [24]. This may indicate a need for further discussion on the development of occupational medicine check-ups and the role of prevention among workers in decreasing absenteeism and presenteeism.

The ailments that respondents work with and that threaten the employee and his/her work is another relevant issue. Headache, fatigue or concentration problems burden the work with a risk of making mistakes, the lack of full engagement or an extension of the time required to perform the task. The aforesaid symptoms were more commonly reported by private sector office workers. It is of great importance to raise the awareness of employees and employers to the fact that it will be more beneficial for all parties if the employee stays at home and rests in order to return to work sooner with full productivity. From the employee’s perspective, it is also more advantageous in terms of possible compensation loss, which would be even greater as a consequence of untreated disease.

From the results of the study discussed, it transpires that, above all, attention should be paid to teachers when analysing sickness presence, especially given that in Poland, there are no databases on chronic diseases and their risk factors split by professional groups, including teachers [25].

## 5. Conclusions

This conducted research shows that there is a necessity for further studies concerning the presence of sick employees in the workplace, especially in terms of the methods for assessing the presenteeism rate. The survey used by the authors was based on employee self-evaluation, which is a subjective assessment and, therefore, has its limitations.

In the researched groups, special attention should be given to teachers and nurses as the sickness presence in these professions may pose a threat to public health. It is suggested to explore this area in future research using a larger study group.

The awareness of sickness presence should be raised both among employees and employers, and the workplace should be a key area. Health education on its consequences, i.e., possible exacerbation of health problems, being a threat to the health of others or ineffective work burdened with the risk of making mistakes, should be propagated.

## Figures and Tables

**Figure 1 healthcare-11-00512-f001:**
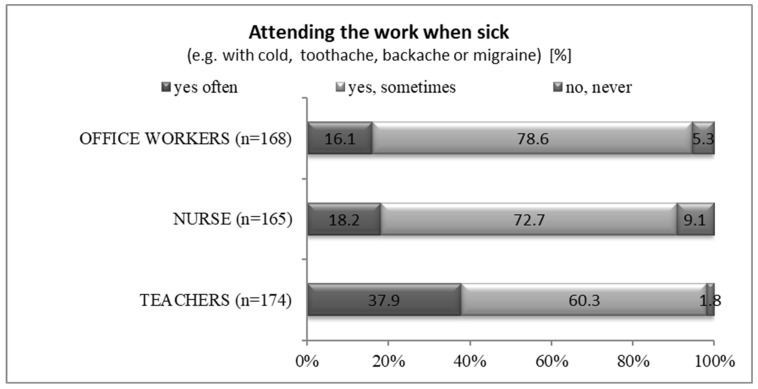
Sickness presence among the respondents.

**Table 1 healthcare-11-00512-t001:** Characteristics of respondents.

		Teachers	Nurses	Office Workers
	TOTAL	N = 174	N = 165	*n* = 168
sex	females	74.1%	94.5%	69.6%
	males	25.9%	5.5%	30.4%
age	mean	43.1	42.9	27.5
	median	42	43	27
	min.	30	30	21
	max.	61	61	42
	standard deviation	7.4	5.9	4.9
type of employment contract	employment contract	100%	100%	32.1%
	commission contract	0.0%	0.0%	64.3%
	contract for a specified task	0.0%	0.0%	3.6%
work schedule	regular	90.9%	54.6%	89.3%
	shift	0.0%	45.5%	0.0%
	irregular	9.1%	0.0%	10.7%
having children	yes	81.0%	45.5%	21.4%
	no	19.0%	54.6%	78.6%
health status	very good	19.0%	36.3%	30.4%
	good	70.7%	27.2%	53.6%
	neither good nor bad	10.3%	27.2%	16.1%
	bad	0.0%	9.1%	0.0%
	very bad	0.0%	0.0%	0.0%
health problems	yes	37.9%	7.3%	17.9%
	no	62.1%	92.7%	82.1%

**Table 2 healthcare-11-00512-t002:** Sickness presence and characteristics of the respondents (%) with *p*-values for the chi-squared test.

		Teachers		Nurses		Office Workers	
		Presence in the Workplace Despite Feeling Unwell, Being Sick or Presenting with Ailments					
		yes	no	yes	no	yes	no
sex	females	100	0.0	90.4	9.6	100	0.0
	males	93.3	6.7	100	0.0	82.4	17.6
*p*-value		0.003		0.329		<0.001	
age group	≤29	0.0	0.0	0.0	0.0	92.7	7.3
	30–39	100	0.0	84.6	15.4	100	0.0
	40–49	100	0.0	100	0.0	100	0.0
	≥50	93.8	6.2	66.7	33.3	0.0	0.0
*p*-value		0.018		<0.001		0.176	
having children	yes	97.9	2.1	80.0	20.0	100	0.0
	no	100	0.0	100	0.0	93.2	6.8
*p*-value		0.400		<0.001		0.107	
health status	very good	90.9	9.1	75.0	25.0	100	0.0
	good	100	0.0	100	0.0	90.0	10.0
	neither good nor bad	100	0.0	100	0.0	100	0.0
	bad	0.0	0.0	100	0.0	0.0	0.0
*p*-value		0.001		<0.001		0.016	
health problems	yes	100	0.0	75.0	25.0	100	0.0
	no	97.2	2.8	92.2	7.8	93.5	6.5
*p*-value		0.182		0.047		0.648	

**Table 3 healthcare-11-00512-t003:** Distribution of responses (%) and *p*-values for the chi-squared test concerning selected ailments that respondents worked with.

	Rhinitis	Sore Throat, Cough	Headache	Increased Temperature	Gastrointestinal Disorders	Toothache	Back/Neck/Joint Pain	Fatigue	Concentration Problems
TEACHERS (*n* = 174)	82.8	87.9	65.5	39.7	24.1	10.3	46.6	60.3	15.5
NURSES (*n* = 165)	45.5	54.6	72.7	18.2	18.2	18.2	27.3	63.6	18.2
OFFICE WORKERS (*n* = 168)	50.0	100	76.8	55.4	7.1	26.8	23.2	76.8	44.6
*p*-value	<0.001	<0.001	0.064	<0.001	<0.001	<0.001	<0.001	0.003	<0.001

**Table 4 healthcare-11-00512-t004:** Distribution of responses (%) and *p*-values for the chi-squared test concerning the reasons for which respondents attended work when sick or presenting with health problems.

	Obligations to Be Met	Lack of a Replacement	Forthcoming Deadline	Workplace Culture	Fear	Loyalty to the Institution
TEACHERS (*n* = 174)	71.9	3.5	47.4	8.8	3.5	31.6
NURSES (*n* = 165)	50.0	40.0	10.0	0.0	40.0	20.0
OFFICE WORKERS (*n* = 168)	39.6	67.9	0.0	5.7	0.0	26.4
*p*-value	<0.001	<0.001	<0.001	0.001	<0.001	0.063

## Data Availability

No additional data are available.

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
