# Peer review of "Sickness Presence among Teachers, Nurses and Private Sector Office Workers"

_healthcare, 2023, doi:10.3390/healthcare11040512_

Round 1

Reviewer 1 Report

Comments attached.

Author Response

We would like to thank the Healthcare for reviewing our article “Sickness presence among teachers, nurses and private sector office workers”. Please find attached the corrected article and cover letter.

Reviewer 2 Report

The topic is interesting, but there are some concerns about the article.

The methodology of the study does not reach the level for publication in the journal. The sample is not representativd, therefore, we can not make a conclusion for the whole population. The authors used t-test as a method, non parametric statistics, which, in my opinion, is not enough to make appropriate conclusions. 

The conclusions should consist the main findings, limitations and suggestions for the future research. 

Author Response

(The authors gave the same response as above.)

Reviewer 3 Report

I congratulate the Authors on an interesting work on the important topic of presenteism. The work, although interesting, has some issues that, in my opinion, should be corrected before publication.

Introduction. In my opinion, this section does not exhaust the subject, what is interesting, the term "presenteism", commonly used in health economics, appears in the work only in the discussion where the understanding of this term in Polish language is questioned. I am not convinced that an international journal is the right place for national semantic discussions. The phenomenon was not properly discussed in the introductory part, without citing the relevant literature on the subject. The work is addressed to a special issue concerning costs, so in my opinion it would be reasonable to refer to the economic consequences here. They appear, in a superficial form, in the discussion.

The material and methods section is unclear.

- there is no reliable description of the statistical methods used.

 - there is no information about the reasons that guided the Authors in choosing professional groups - why them?

Discussion section. As mentioned above, I have serious doubts about the value of semantic considerations. The authors indicate that the phenomenon they are discussing is difficult to measure, but at the same time, they do not say anything about research tools dedicated to this purpose, and they compare own the results to the results of other authors only in a limited way.

Author Response

(The authors gave the same response as above.)

Round 2

Reviewer 2 Report

The paper can be accepted.

Reviewer 3 Report

Thank you for making changes to the article. I accept it in its current form and warmly congratulate you on an interesting study.